# Characterisation and modelling of potassium-ion batteries

Shobhan Dhir [1], John Cattermull [1,2], Ben Jagger [1], Maximilian Schart[1], Lorenz F. Olbrich [1], Yifan Chen [1], Junyi Zhao[1], Krishnakanth Sada [1], Andrew Goodwin [2] & Mauro Pasta [1] ✉

Potassium-ion batteries (KIBs) are emerging as a promising alternative technology to lithium-ion batteries (LIBs) due to their significantly reduced dependency on critical minerals. KIBs may also present an opportunity for superior fast-charging compared to LIBs, with significantly faster K-ion electrolyte transport properties already demonstrated. In the absence of a viable K-ion electrolyte, a full-cell KIB rate model in commercial cell formats is required to determine the fast-charging potential for KIBs. However, a thorough and accurate characterisation of the critical electrode material properties determining rate performance—the solid state diffusivity and exchange current density—has not yet been conducted for the leading KIB electrode materials. Here, we accurately characterise the effective solid state diffusivities and exchange current densities of the graphite negative electrode and potassium manganese hexacyanoferrate $K_2Mn[Fe(CN)_6]$ (KMF) positive electrode, through a combination of optimised material design and state-of-the-art analysis. Finally, we present a Doyle-Fuller-Newman model of a KIB full cell with realistic geometry and loadings, identifying the critical materials properties that limit their rate capability.

Batteries are critical for decarbonisation of the transport sector and energy storage for renewables. However, the leading lithium-ion (Li-ion) chemistries meeting this demand are highly intensive in terms of critical minerals including lithium, nickel, cobalt, graphite and copper[1], many of which have experienced exceptional price volatility over recent years, posing significant uncertainty for their future security of supply[2–4]. Therefore, the case for alternative chemistries which can fulfil some lithium-ion battery (LIB) functions with reduced critical mineral dependency is significantly growing[5–9]. One of the most promising positive electrode materials for potassium-ion batteries (KIBs), the potassium manganese hexacyanoferrate $K_2Mn[Fe(CN)_6]$ (KMF), contains no critical minerals while K-ion can also utilise aluminium negative electrode current collectors unlike Li-ion, removing the need for any copper in the cell[5]. KIBs also present a significant advantage over sodium-ion batteries (NIBs) as $K^+$ can intercalate into the graphite

electrodes used in LIBs[10,11]. Therefore, one of the primary components of KIBs is already available at global industrial scale, unlike for NIBs[5].

Fast electric vehicle (EV) battery charging rates (-4 C[12]) are also becoming increasingly important for consumers, however, LIBs are limited in their capability for fast-charging[13]. Critical challenges limiting accessible capacities at high rates in LIBs include slow electrolyte transport, Li metal plating and constant solid electrolyte interphase (SEI) formation[12–14]. KIBs, however, may present an advantage over LIBs in terms of fast-charging. We recently demonstrated that the K-ion electrolyte potassium bis(fluorosulfonyl)imide (KFSI) in 1,2-dimethoxyethane (DME) displays significantly higher salt diffusivities and cation transference numbers than the Li-ion equivalent, resulting in reduced electrolyte concentration gradient formation, thus faster electrolyte transport and lower electrolyte concentration overpotentials at higher charging rates[15]. This is due to the larger size of $K^+$,

[1]Department of Materials, University of Oxford, Oxford OX1 3PH, UK. [2]Inorganic Chemistry Laboratory, Department of Chemistry, University of Oxford, Oxford OX1 3PH, UK. ✉e-mail: mauro.pasta@materials.ox.ac.uk

resulting in a lower charge density and weaker interactions with solvent molecules[15].

In the absence of an electrolyte capable to provide both a stable SEI for the graphite negative electrode and practical coulombic efficiencies at the high operating voltages of the leading positive electrodes[5], the experimental validation of the rate capability of KIBs is not currently possible. Therefore, to understand the potential of K-ion fast-charging, and invigorate the search for a suitable electrolyte, requires full-cell Doyle-Fuller-Newman (DFN)[16,17] modelling of K-ion in a commercial cell format. This requires characterisation of the KIB critical electrode material properties which also significantly contribute to determining rate performance, the solid-state diffusivity, $D$, and the exchange current density, $j_0$. These properties have been characterised, estimated and parameterised in teardown analyses for commercial Li-ion cells[18–23] but not yet for K-ion.

Accurate characterisation of $D$ is particularly challenging. Typically measured using galvanostatic intermittent titration technique (GITT) or the potentiostatic intermittent titration technique (PITT), $D$ characterised by these methods can vary by several orders of magnitude even for the same material[24,25]. This is a result of a variety of sources of error including unsuitable experimental conditions, inaccurate analysis and high uncertainties of critical parameters. Both techniques were originally developed for dense, single-phase, planar bulk materials[26,27]. However, leading battery electrode materials today are multi-particle, often multiphase, porous materials, which poses significant additional challenges.

There is considerable debate regarding the $D$ measured in multiphase systems by techniques such as GITT and PITT[28–33]. However, Ceder et al. showed that GITT and PITT are still accurate in measuring $D$ in multiphase systems described with phase field modelling accounting for the phase changes[29]. In two-phase regions, the measured $D$ is considered an effective diffusivity ($\widetilde{D}$) rather than a chemical diffusivity with contributions from the chemical diffusivities of the two stable phases as well as the movement of the interphase boundary[29,30]. PITT has been found to be inferior to GITT for multiphase systems due to the insufficiently low potential step that can be applied in the two-phase potential plateaus and the inability to apply sufficient integration time[28,29]. PITT is also inherently limited compared to GITT due to there being no zero-current relaxation periods to separate current-related overpotentials[25].

Kang and Chueh recently conducted a systematic analysis of the sources of error in GITT application for battery materials, providing recommendations for improved GITT experimental conditions and analysis, as well as determining an optimised modified fitting method for more accurate determination of $D$ (method denoted herein as Kang-Chueh GITT). In the Kang-Chueh GITT analysis ('Methods') $D$ [m² s⁻¹] is calculated according to Eq. (1)[25,34]:

$$D = \frac{4}{\pi} \left( \frac{I V_m}{zFS} \right)^2 \left[ \left( \frac{\partial V_{eq}}{\partial x} \right) / \left( \frac{dV}{d(\sqrt{t_{relax} + \tau} - \sqrt{t_{relax}})} \right) \right]^2 \quad (1)$$

where $I$ [A] is applied current, $V_m$ [m³ mol⁻¹] is molar volume, $F$ [A s mol⁻¹] is the Faraday constant, $z$ [–] is the charge number, $S$ [m²] is the electrochemically active surface area, $\frac{\partial V_{eq}}{\partial x}$ [V] is the derivative of the Nernst voltage with stoichiometry, $V$ [V] is voltage during relaxation, $t_{relax}$ [s] is relaxation time and $\tau$ [s] is pulse duration.

Important sources of error in conventional GITT include composition-dependent overpotentials during conventional pulse analysis, finite-size effects due to inappropriate pulse conditions or particle size, convolution of electrolyte transport limitation with $D$, or counter electrode overpotential contributions in two-electrode cells[25]. Utilising accurate relaxation-only analysis, large particles and appropriate pulse conditions to minimise finite-size effects, low sample mass loading with high porosity to ensure that the potential relaxation

profile is governed by $D$, and three-electrode cells, mitigate these key sources of error[24,25]. However, one of the most critical sources of uncertainty for both GITT and PITT applied to porous electrode materials is the determination of the electrochemically active surface area, $S$, or the diffusion length, $L$[24,25,31,35]. With an inverse-square relation, the resulting $D$ is highly sensitive to this parameter (Eq. (1)). This ambiguity in $S$ results in orders of magnitude difference in $D$ alone. Therefore, the accuracy of $D$ could be substantially improved with the design of a material with greater morphological homogeneity.

Reaction kinetics at the electrode-electrolyte interface for porous electrode materials are conventionally taken to follow the Butler-Volmer kinetic laws. The kinetic reaction rate is governed by the exchange current density $j_0$ according to the Butler-Volmer equation ('Methods')[17]. By measuring the charge-transfer resistance, $R_{ct}$ [Ω], using electrochemical impedance spectroscopy (EIS), $j_0$ [A cm⁻²] can be determined through linearising the Butler-Volmer equation in combination with $S$ [cm²], the Faraday contant, $F$ [A s mol⁻¹], the molar gas constant, $R$ [ J mol⁻¹ K⁻¹], and temperature, $T$ [K] (Eq. (2)). By measuring $j_0$ at various stoichiometries, the plot of $j_0$ over composition can be fitted to a form of the Butler-Volmer equation (Eq. (3), 'Methods') to determine the constant reference current for the reaction, $k_0$, which can be utilised in the DFN model[17,18,20,36]. To isolate the $R_{ct}$ and hence $j_0$ at a single electrode requires the use of three-electrode cells[17]. There are modifications to classical Butler-Volmer kinetics for multiphase materials that have been proposed recently from the work of Bazant et al. in multiphase porous electrode theory (MPET)[37] or coupled ion-electron theory (CIET)[38]. However, both models predict the exchange current densities for lithium iron phosphate (LFP) and graphite comparably to the values obtained using the classical Butler-Volmer model[39–41]. Therefore, similar to $D$, one of the most significant uncertainties in determining $j_0$ is also $S$, though it is less sensitive than $D$ with an inverse rather than an inverse-square relation (Eq. (2)). Hence, again, improved homogeneous material design would improve the determination of $j_0$.

$$j_0 = \frac{RT}{SFR_{ct}} \quad (2)$$

Therefore, in this study we characterise the effective solid state diffusivities and exchange current densities of the leading K-ion electrode materials: the graphite negative electrode and $K_2Mn[Fe(CN)_6]$ (KMF) positive electrode (Fig. 1). To mitigate the critical source of uncertainty and error of $S$, for the determination of $D$ and $j_0$, we synthesise a highly homogeneous and monodisperse KMF positive electrode material enabling a considerably more accurate determination of $S$, while we analyse the active area of a commercial synthetic graphite. To determine the effective $\widetilde{D}$ of these materials we employ the state-of-the-art Kang-Chueh GITT technique and analysis to mitigate common errors from conventional GITT application[25,34]. We determine $j_0$ for both materials using EIS. KFSI in triethyl phosphate (TEP) was utilised as the electrolyte as it has arguably achieved the best K-ion full-cell performance[42]. Finally, we present a Doyle-Fuller-Newman model of a KIB full cell in a hypothetical cell based on the commercial LG M50 cylindrical cell format, enabling us to identify the critical limitations in realising fast-charging KIBs.

## Results
### Graphite negative electrode
High-crystallinity synthetic graphite exhibits superior capacity retention to low-crystallinity graphite in K-ion cells[43]. Therefore, a commercial, highly crystalline, synthetic graphite was utilised (SGP5, SEC Carbon). A larger particle size was utilised to minimise finite-size effects in the Kang-Chueh GITT measurements[25]. Pawley refinement of synchrotron X-ray diffraction (XRD) data confirms the highly crystalline, phase-pure nature of the synthetic graphite (Fig. 2a). Graphite

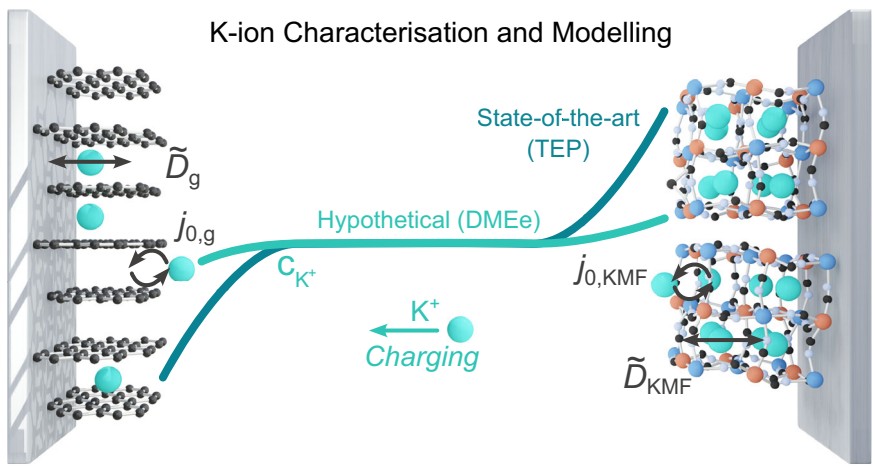

**Fig. 1 | K-ion characterisation and modelling.** Schematic of the leading K-ion chemistry characterised and modelled. The graphite negative electrode (left) and the potassium manganese hexacyanoferrate (KMF) positive electrode (right). The effective solid-state diffusivities, $\widetilde{D}_i$, and exchange current densities, $j_{0,i}$, were characterised here, enabling full-cell Doyle-Fuller-Newman modelling in combination with electrolyte transport properties of the current state-of-the-art K-ion electrolyte in the K-ion research community KFSI:TEP (TEP) or a hypothetical electrolyte with equivalent transport properties of KFSI:DME (DMEe) electrolyte characterised previously[15].

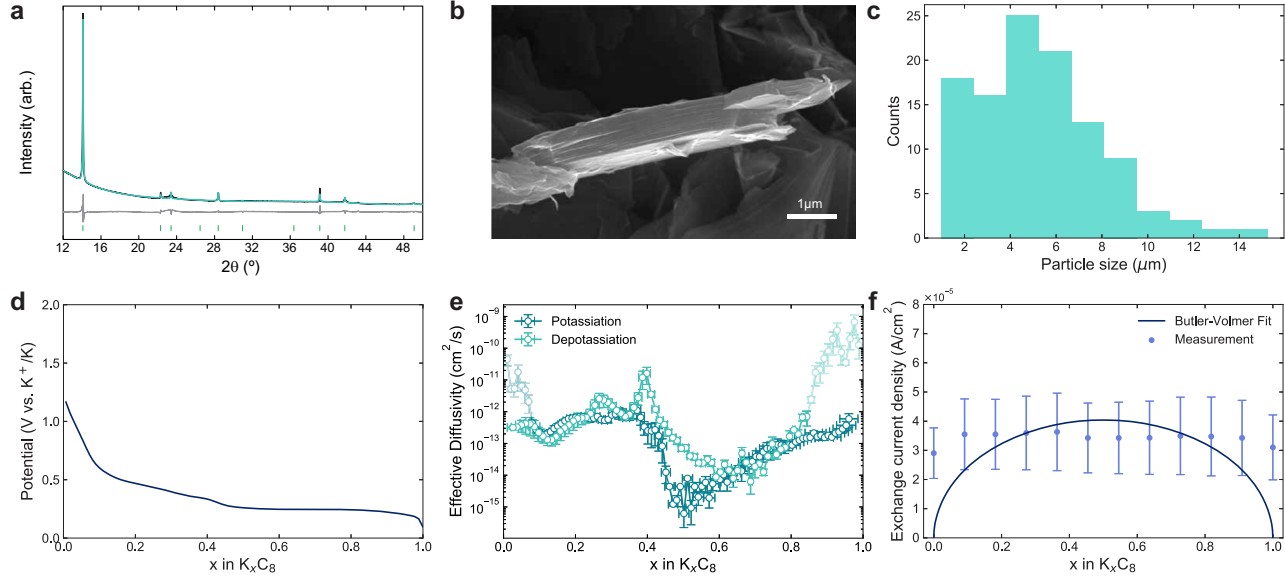

**Fig. 2 | K graphite characterisation.** Material and kinetic characterisation of the potassium synthetic graphite. **a** Pawley refinement of synchrotron XRD pattern (data in black, fit in teal, data-fit difference in grey, and reflection positions below in green). **b** SEM image of a single particle. **c** Particle size distribution. **d** Open-circuit voltage (OCV) profile from the Kang-Chueh GITT during depotassiation at 20 °C. **e** Effective diffusivity over composition from Kang-Chueh GITT analysis at 20 °C. Greyed out points indicate data which may be affected by SEI formation. **f** Exchange current density over composition at 20 °C. Error bars depict the standard error in the mean from at least two repeat measurements.

particles comprise of highly oriented layers, shown in a single graphite particle in the the SEM image in Fig. 2b, where the basal planes are parallel to the graphene layers while the edge planes expose the end faces of the graphene layers[44]. Intercalation of $K^+$ into graphite is similar to $Li^+$ and only occurs through these edge planes where graphene layers are exposed as shown clearly in Fig. 2b, with the diffusion process advancing into the particle centre along the basal plane[44–47]. Therefore, the graphite particle $S$ is only this edge area (Fig. 2b). Supplementary Fig. 1a and b show other SEM images of the graphite particles in powder and in a cast electrode, respectively, showing the flake-like morphology expected. The graphite average particle size distribution from SEM analysis is shown in Fig. 2c, with an average graphite particle size of 5.28 μm and thickness of 0.17 μm (Supplementary Fig. 2).

Based on the active edge area and the SEM average particle size and thickness geometric analysis, the graphite particles were approximated as discs (Supplementary Fig. 3) with $S$ determined from this disc edge geometric shaded area where $K^+$ intercalation occurs. Levi et al. and Yang et al. also accounted for this edge intercalation area in graphite in their determination of Li graphite $\widetilde{D}$[45,46]. Adsorption methods such as Brunauer-Emmett-Teller (BET) are most commonly used to characterise $S$. However, in addition to being a limited and inaccurate proxy for $S$, for instance being unclear whether very fine pores measured by BET are wetted by the electrolyte[25,48], it is highly unsuitable for determining $S$ of graphite as it would include the large area of inactive basal plane regions of the graphite particles. The graphite reversible capacity is close to the theoretical capacity (260 vs. 279 mA h g$^{-1}$, respectively, Supplementary Fig. 4), this indicates there is

very little inactive region of the cast graphite electrode, supporting our method of estimation of $S$.

Figure 2d shows the open-circuit voltage (OCV) profile of the graphite and Fig. 2e shows the effective $\widetilde{D}_g$ results over stoichiometry from the Kang-Chueh GITT and determined $S$. The results show the median $\widetilde{D}_g$ from both potassiation and depotassiation is $2.32 \times 10^{-13}$ cm² s⁻¹. This appears to be two to three orders of magnitude lower than the values reported for Li⁺ in graphite[18,36,45], suggesting slower diffusion for K graphite. However, the Li⁺ $\widetilde{D}_g$ should be measured using the Kang-Chueh GITT for a more meaningful comparison.

Studies have shown using operando XRD and Raman[49,50] that K⁺ intercalation into graphite progresses through several two-phase transformations and Onuma et al. proposed the following staging evolution[49]:

$$\text{Graphite} \xrightarrow{} \begin{array}{c}\text{Disorderly stacked high stage}\\\text{Graphite} - KC_{96}\end{array} \xrightarrow{} \begin{array}{c}\text{Stage 4L} - 3L\\KC_{96} - KC_{24}\end{array} \xrightarrow{} \begin{array}{c}\text{Stage 2L}\\KC_{28} - KC_{24}\end{array} \xrightarrow{} \begin{array}{c}\text{Stage 1}\\KC_8\end{array}$$

Stages 4L, 3L and 2L exhibit "liquid-like" in-plane potassium distributions and Daumas-Hérold defects are generated during the phase transformations. The phases therefore have variable compositions and a high concentration of defects, causing their potentials to change during intercalation and explaining the lack of clear plateaus in Fig. 2d for $x < 0.4$. In contrast, stage 1 forms with a fixed composition through the complete filling of the graphene layers with potassium, eliminating Daumas-Hérold defects and resulting in a constant potential, as evident by the flat voltage plateau for $x \sim 0.4$ to 0.8 in Fig. 2d as $KC_8$ forms. This appears to coincide with a two to three orders of magnitude drop in the effective $\widetilde{D}_g$ between $x \sim 0.4$ to 0.8 (Fig. 2e). However, we exercise caution in analysing regions where $\frac{\partial V_{eq}}{\partial x}$ approaches zero (Supplementary Fig. 5), which in turn leads to $\widetilde{D} \sim$ zero (Eq. (1)), which is not physical[31].

Onuma et al. further observed hysteresis between the intercalation and deintercalation processes, with a stage 2 ($KC_{16}$) structure able to form from $KC_8$ initially before Daumas-Hérold defects are again necessary for further deintercalation[49]. These differences may explain some of the directional differences in the $\widetilde{D}_g$ profile during potassiation and depotassiation, particularly since composition-dependent overpotentials are avoided here through the relaxation-only GITT analysis[25]. However, autocatalysis effects may also contribute to direction-dependencies; therefore it is advisable to refrain from assigning excessive physical meaning to the direction-dependency[25].

We further note that $\widetilde{D}_g$ appears to be significantly greater at low $x$ during potassiation and at high $x$ during depotassiation, which is not expected physically. Both of these inflated regions correspond to the first few pulses after switching the current direction, and may therefore be caused by SEI formation. Although the GITT measurements were performed with the current leading K-ion electrolyte (KFSI:TEP), continuous SEI formation is still a common issue, even after numerous cycles[42,51]. A recent study further evidences that the SEI is partially soluble[52] and its composition can change dynamically during cycling[14,53]. Therefore, there may be a restructuring of the SEI that takes place when the direction of the current pulse is changed, consuming capacity until a sufficiently passivating structure is formed. This is supported by evidence that the SEI composition on graphite in K-ion cells changes considerably between charge and discharge[54]. The necessarily short current pulses applied here mean that this may influence the $\widetilde{D}_g$ results over several pulses. We therefore believe the potassiation data to give a more reliable $\widetilde{D}_g$ for $x \sim 1$, and depotassiation for $x \sim 0$, as highlighted in Fig. 2e.

The impact of finite-size effects on the Kang-Chueh GITT $\widetilde{D}_g$ can be assessed through evaluating the dimensionless pulse time ($\hat{\tau} = \widetilde{D}\tau/L^2$)[25]. Supplementary Fig. 6 shows the dimensionless pulse time for the majority of the graphite Kang-Chueh GITT data is within the ideal valid semi-infinite region for 3D geometries, minimising finite-size

effects[25,34]. Diffusion in graphite particles can also be considered 2D along the graphene planes, providing further mitigation against finite-size effects. Supplementary Fig. 7 shows the $\widetilde{D}_g$ evaluated using PITT agrees reasonably well with the Kang-Chueh GITT results (Supplementary Note 1), though with less sensitivity to composition as expected from the limitations of potential step size in plateau regions, as mentioned previously. However, PITT results are inherently limited compared to relaxation-only Kang-Chueh GITT as described in the introduction and Supplementary Note 1. The PITT $\widetilde{D}_g$ minima are shallower than for Kang-Chueh GITT, matching the findings from Markevich et al. who found that PITT is more susceptible to parasitic current contributions than GITT, resulting in overestimated and less accurate $D$[28].

Figure 2f shows $j_{0,g}$ of the graphite over stoichiometry in 2 m KFSI:TEP electrolyte fitted to the Butler-Volmer equation (Eq. (3)). The mean $j_{0,g}$ over the composition is $3.42 \times 10^{-5}$ A cm⁻². $j_{0,g}$ is similar to that found for Li graphite[18,20] indicating similar charge-transfer reaction kinetics between Li⁺ and K⁺ and graphite. A critical challenge with accurately determining $R_{ct}$ from EIS is the fact that constantly evolving SEI and passivation layer formation occur at similar frequency ranges to charge transfer[55]. The equivalent circuit used to determine $R_{ct}$ and an example impedance spectrum for K graphite are shown in Supplementary Figs. 8 and 9, respectively. $R_{ct}$ was represented by R2 in the equivalent circuit, with R1 representing the SEI due to evidence that SEI impedance has a higher characteristic frequency than $R_{ct}$[56,57]. From the Butler-Volmer fit (Eq. (3)) the reference current for the reaction $k_{0,g}$ is $8.07 \times 10^{-5}$ A cm⁻². The poor Butler-Volmer fit for the graphite $j_{0,g}$ matches the findings of Ecker et al. and Schmalstieg et al. for Li graphite[18,58], though O'Regan et al. achieved a good fit for Li graphite[20]. These results indicate that the SEI interferes with the impedance measurements for the graphite electrode. Overlapping time constants for SEI formation and charge-transfer at the graphite electrode may make it difficult to correctly isolate $R_{ct}$ and thus very accurately determine $j_{0,g}$, however, this provides a reasonable order of magnitude for $j_{0,g}$, as required for the model.

## Potassium manganese hexacyanoferrate positive electrode

Figure 3a and b show the large, highly crystalline, non-agglomerated and cuboid KMF material—synthesised via a citrate-assisted co-precipitation ('Methods'). Supplementary Fig. 1c and d show the material is homogeneous and monodisperse. Synchrotron XRD measurement of the KMF sample confirmed that the high degree of crystallinity achieved through more traditional synthesis had been retained (Fig. 3a). Further, Rietveld refinement revealed that a near identical structure and higher potassium concentration (1.871(3) per formula unit) compared to previous studies was produced (Supplementary Note 2)[5,59,60]. Elemental analysis by inductively coupled plasma mass spectrometry (ICP-MS) also indicated a low-vacancy/high potassium content from the Fe:Mn ratio of 0.98(5), giving a chemical formula of $K_{1.871(3)}Mn[Fe(CN)_6]_{0.98(5)}$ from the combined XRD/ICP-MS analysis. The material was synthesised to have as large particles as possible while maintaining performance to minimise finite-size effects and ensure $D$ limitation in the Kang-Chueh GITT measurements. From SEM analysis the average KMF particle size was identified as 1.02 μm (Fig. 3c), significantly larger than other KMF materials synthesised[42,59].

There has been a historical mischaracterisation of $\widetilde{D}$ for Prussian blue analogue (PBA) materials, with many studies characterising in the order of $10^{-8}$ to $10^{-11}$ cm² s⁻¹[35]. This is due to several sources of error. First, as PBAs are frequently synthesised as agglomerated nanoparticles, $S$ or the diffusion length, $L$, ($L \propto \frac{nV_m}{S}$ for a cube, where $n$ is the number of moles) is often mischaracterised based on the agglomerate rather than the primary particle size (or for electrodeposited PBA films the film thickness rather than the individual nanoparticle size) resulting in significant overestimates of $\widetilde{D}$ since $\widetilde{D}$ is inversely proportional to $S^2$ or proportional to $L^2$[35]. Inhomogeneous material, poor

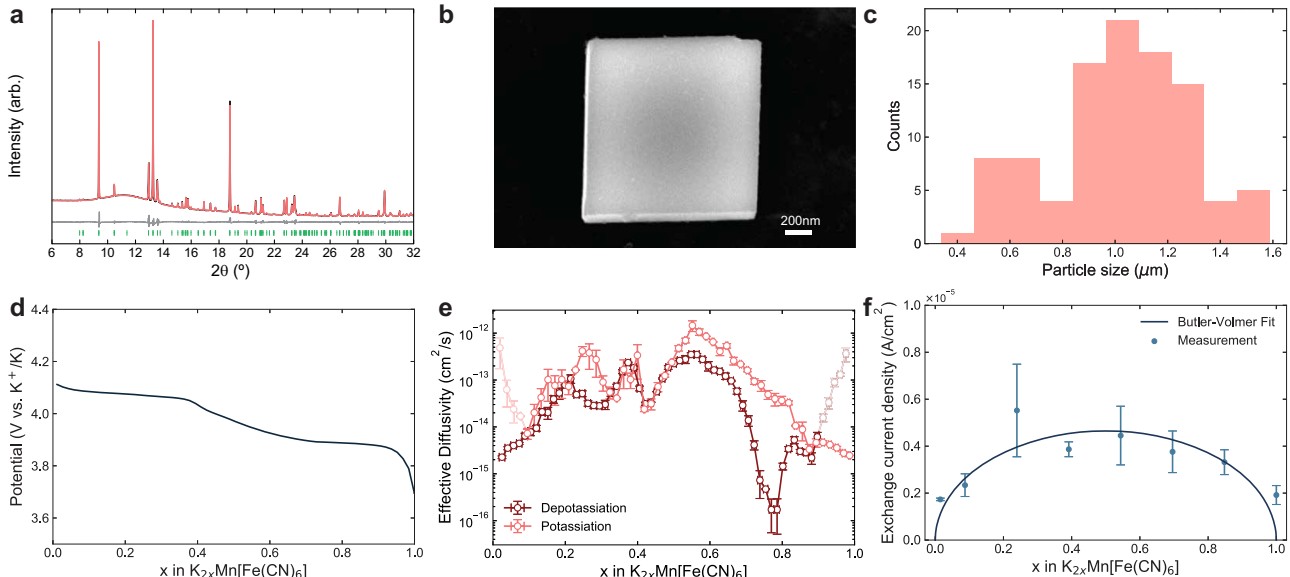

**Fig. 3 | KMF characterisation.** Characterisation of the potassium manganese hexacyanoferrate $K_2Mn[Fe(CN)_6]$ (KMF). **a** Rietveld refinement of synchrotron XRD pattern (data in black, fit in peach, data-fit difference in grey, and reflection positions below in green). **b** SEM image of a particle. **c** Particle size distribution. **d** Open-circuit voltage (OCV) profile from Kang-Chueh GITT during depotassiation at 20 °C. **e** Effective diffusivity over composition from Kang-Chueh GITT at 20 °C. Greyed out points indicate data which may be affected by CEI formation. **f** Exchange current density over composition at 20 °C. Error bars depict the standard error in the mean from at least two repeat measurements.

morphological characterisation and poor diffusivity analysis have also led to mischaracterisation[35]. However, recently Komayko et al. conducted a detailed analysis of several PBA materials[35] using PITT analysis and improved characterisation of the materials, finding $\widetilde{D}$ of various PBA materials assessed was in fact around four orders of magnitude lower than conventionally measured and of the order $10^{-12}$ to $10^{-15}$ cm² s$^{-1}$ [35]. However, they did not characterise $\widetilde{D}$ for the KMF material.

The highly uniform morphology and lack of agglomeration of the KMF synthesised here enables significantly increased accuracy in the determination of $S$, which has been a significant source of error in the determination of $\widetilde{D}$ in the past[35]. Due to this morphological homogeneity, and the 3D framework structure of PBAs enabling K$^+$ insertion through all exposed facets[61], the geometric surface area of the KMF particle is a good measure of $S$. Given the reversible capacity is very close to the theoretical capacity (141 vs. 155 mA h g$^{-1}$, Supplementary Fig. 10), this indicates there is very little inactive region of the cast KMF electrode, supporting our method of estimation of $S$.

The OCV profile in Fig. 3d shows the two well-defined plateaus, which are characteristic of the low-vacancy KMF. Multiphase behaviour in PBA-positive electrodes is a consequence of maximising the theoretical capacity by reducing the vacancy content, which induces highly correlated distortions in the structure[61]. From previous in situ XRD studies we understand there to be three dominant phases; the Jahn-Teller distorted ($x = 0$) and slide distorted ($x = 1$) phases with an apparently undistorted phase at the intermediate $x = 0.5$ composition[62]. We rationalise results from the Kang-Chueh GITT experiment in this context. The results from the Kang-Chueh GITT (Fig. 3e) give a median $\widetilde{D}_{KMF}$ from both potassiation and depotassiation of $5.50 \times 10^{-14}$ cm² s$^{-1}$, showing $\widetilde{D}_{KMF}$ is ~ four times lower than the $\widetilde{D}_g$, indicating slower K$^+$ transport in the KMF. The drops in $\widetilde{D}_{KMF}$ appear to align with the two plateaus between $x$ ~ 0.95 to 0.75 and $x$ ~ 0.4 to 0.1, the former of which corresponds to an apparent drop of three to four orders of magnitude in the effective $\widetilde{D}_{KMF}$. However, we once again exercise caution in analysing regions where $\frac{\partial V_{eq}}{\partial x}$ approaches zero (Supplementary Fig. 11), and without reliable quantification of the phase behaviour from in situ structural techniques one cannot comment further on this result.

The $\widetilde{D}_{KMF}$ results obtained here are within the lower range identified by Komayko et al. for other PBA materials evaluated with improved materials and analysis[35], thus supporting the accuracy of the $\widetilde{D}_{KMF}$ results ascertained here. Using the Kang-Chueh GITT relaxation analysis also avoids the current-related overpotential errors from PITT analysis in their study[25,35]. Authors He and Nazar found that the KMF analogue $K_2Fe[Fe(CN)_6]$ (KFF) displayed notably inferior rate capability to its sodium equivalent when their crystallites are micron-sized[63]. This suggests that K$^+$ diffusion is slower for K-PBA materials compared with Na equivalents.

Similar to the graphite case, $\widetilde{D}_{KMF}$ appears to also be enhanced for the first few pulses after the direction of the current pulse is changed, which could also indicate regions where cathode electrolyte interphase (CEI) formation is influencing the results. We therefore again believe the potassiation data to give a more reliable $\widetilde{D}_{PBA}$ for $x$ ~ 1, and depotassiation for $x$ ~ 0, as highlighted in Fig. 3e. Supplementary Fig. 12 assesses the dimensionless pulse time for the KMF, also showing the majority of data is within or close to the ideal valid semi-infinite region for 3D geometries, again minimising finite-size effects[25,34]. Supplementary Fig. 13 shows the PITT $\widetilde{D}_{KMF}$ results, again showing they match the Kang-Chueh GITT results in the general trend and average magnitude, however, again with limited composition resolution in the two-phase regions as identified in other works[28,29].

Figure 3f shows the KMF exchange current density, $j_{0,KMF}$, over stoichiometry in 2 m KFSI TEP electrolyte, measured in a three-electrode cell, and again fitted to a form of the Butler-Volmer (Eq. (3) and 'Methods'). The data fits the Butler-Volmer trend well, matching good Butler-Volmer $j_0$ fits found for Li-ion lithium nickel manganese cobalt oxide (NMC) materials[18,58]. For the fits the charge-transfer coefficients ($\alpha_a$ and $\alpha_c$) are maintained as 0.5, as is conventionally assumed[17,18,58]. O'Regan et al. allow $\alpha_a$ and $\alpha_c$ to be a free fitting parameter to improve the Butler-Volmer fit[20]. However, since $\alpha_a$ and $\alpha_c$ are highly difficult to measure accurately and validate[17], and as 0.5 gave a good fit for the KMF, this conventional assumption was maintained. From the Butler-Volmer fit (Eq. (3)) $k_{0,KMF}$ was determined as $0.93 \times 10^{-5}$ A cm$^{-2}$. The equivalent circuit used to determine the KMF $R_{ct}$ and an example KMF impedance spectrum are shown in

Supplementary Fig. 14 and 15, respectively. Again, $R_{ct}$ is represented by R2 in the equivalent circuit as R2 results in $j_{0,KMF}$ clearly fitting the Butler-Volmer trend, and also due to evidence that passivation layer formation occurs at higher frequencies than $R_{ct}$[57]. The results show a mean $j_{0,KMF}$ of $0.34 \times 10^{-5}$ A cm$^{-2}$ over the composition which is approximately an order of magnitude lower than $j_{0,g}$, indicating notably faster kinetics for the graphite negative electrode.

The sluggish $j_{0,KMF}$ charge-transfer kinetics identified could finally explain the poor rate capability found by other studies assessing the KMF electrochemical performance[59,63], where the underlying cause had not been identified[5]. For context $j_{0,KMF}$ is almost two orders of magnitude lower than that found for NMC positive electrode materials indicating significantly less efficient reaction kinetics to the high energy density Li-ion metal oxides[18,20,58]. However, the $j_{0,KMF}$ appears around half an order of magnitude higher than LFP (from $4.7-16.7 \times 10^{-7}$ A cm$^{-2}$) determined from fitting to commercial LFP electrode experimental data[23] and to other models calibrated against experimental LFP testing[64]. Therefore, the kinetics appear to be similar or slightly better than that for LFP. Given LFP Li-ion would be the competitor for K-ion, rather than the high energy density high nickel positive electrodes[5], the comparable $j_0$ for KMF to LFP is promising for the competitiveness of K-ion.

## Full-cell potassium-ion modelling

To understand the potential of KIBs for fast-charging, we developed a KIB full-cell DFN model in a hypothetical cell based on the commercial LG M50 cylindrical cell format[20,36], ensuring realistic electrode thicknesses, parameters and loadings were used. As mentioned, since there is no current K-ion electrolyte which provides practical coulombic efficiencies, two K-ion cells were modelled with alternative electrolytes. First, using KFSI:TEP, which is considered the current leading K-ion electrolyte in the K-ion research community, and is also the electrolyte we used in our experimental investigation[42,65]. Second, using a hypothetical electrolyte with equivalent properties to KFSI:DME (DMEe), the only other non-aqueous K-ion electrolyte fully characterised until now[15]. KFSI:DME properties were used as a model electrolyte to indicate full-cell K-ion performance once a potential suitable electrolyte has been developed and optimised. It is important to note KFSI:DME will not be the electrolyte utilised in commercial KIBs unless additives are developed which mitigate its tendency for cointercalation into graphite[5]. The K-ion models were developed using the electrode properties characterised here ($\widetilde{D}_g$, $j_{0,g}$, $\widetilde{D}_{KMF}$ and $j_{0,KMF}$) in combination with the KFSI:TEP electrolyte properties characterised by Zhao et al. (Supplementary Note 3 and Supplementary Table 3)[66] or the KFSI:DME transport properties fully characterised in our previous work (Supplementary Note 3)[15]. The cells were modelled using the battery modelling package PyBaMM[67]. The total energy of each K-ion cell was simulated to be 7.3 Wh by adjusting the electrode thicknesses. The half-cell OCV profiles for the graphite negative electrode and KMF positive electrode determined here were implemented in the model (Figs. 2d and 3d, respectively). All electrode properties, including positive and negative electrode material particle size and electrode porosities were kept constant for both cases, and the negative/positive electrode capacity ratios (NP ratios) were set to be 1.1 as typical in commercial Li-ion cells[68]. Full details of the model and parameters are described in the 'Methods' and Supplementary Table 3.

The median $\widetilde{D}_g$ and $\widetilde{D}_{KMF}$ (Supplementary Figs. 16 and 17) are being used in the model to represent the Kang-Chueh GITT data characterised in the regions where potassiation and depotassiation match, while minimising exposure to the extreme values—the high values where SEI/CEI formation and finite-size effects are likely to have some impact due to being above the ideal semi-infinite regime (Supplementary Figs. 6 and 12), and also avoiding impact from the potential plateau regions when $\frac{\partial V_{eq}}{\partial x}$ approaches zero (Supplementary Figs. 5 and 11) as mentioned previously.

In commercial LFP LIBs, LFP particles need to be nanosized to increase the surface area available for reaction and decrease the diffusion length due to the substantially lower $j_0$ and $D$ of LFP than Li graphite. In the K-ion cell, $\widetilde{D}_{KMF}$ is approximately four times lower than $\widetilde{D}_g$, and $j_{0,KMF}$ is around one order of magnitude lower than $j_{0,g}$. Consequently, the KMF particles must also be nanosized to match the faster kinetics of the graphite negative electrode. In the model, the KMF particle sizes are set to 500 nm, consistent with the commercial LFP particle size[69].

Figure 4a depicts the K-ion cell being modelled with the two electrolytes. The KMF positive electrode is 66% thicker than the graphite negative electrode due to the lower capacity and bulk density of the KMF material[5]. Figure 4b shows the specific energy and energy density of the K-ion chemistry based on the positive and negative electrode theoretical capacities and the simulated galvanostatic profile (Fig. 4c) using the stack-level model developed previously[5,70]. Figure 4c displays the DFN simulated galvanostatic profiles of the two chemistries during a 1 C charge, demonstrating the higher overpotentials experienced by the TEP cell. Finally, Fig. 4d shows the fast-charging performance comparison for the two K-ion chemistries, demonstrating the DMEe shows significantly higher rate capability than the TEP cell, achieving significantly higher accessible capacities at all rates simulated. Even at the fast-charging rate of 5 C the DMEe cell can access 34% capacity compared to 8% for the TEP cell. To understand the reasons for the significant difference in rate capability, Fig. 4e and f plot the overpotential components for each K-ion chemistry during a 5 C charge until the upper cut-off voltage is reached, beyond which cell degradation processes may take place[13]. Figure 4e shows the electrolyte concentration overpotentials are exceptionally high and growing quickly for the TEP K-ion cell early in the charge cycle at SOC < 0.1, causing the upper-cutoff voltage to be quickly reached. This is the result of significant electrolyte concentration gradient formation in the TEP cell due to its lower salt diffusivity (~ one order of magnitude lower than DMEe (Supplementary Table 3)[15], thus limiting the transport of K$^+$ to the graphite negative electrode during charge. The ionic conductivity of the TEP electrolyte is also ~ five times lower than that of the DMEe electrolyte (Supplementary Table 3)[15], resulting in larger electrolyte ohmic overpotentials. The significantly slower electrolyte transport properties are a result of highly viscous nature of the TEP electrolyte (for context ~ five times higher viscosity than a commercial Li-ion carbonate electrolyte LP30[51,71]). Therefore, though KFSI:TEP is the current leading electrolyte for the K-ion research community using low loading in coin cells, this electrolyte is unsuitable for even moderately high charge rates using commercial cell electrode thicknesses and loading, and an electrolyte with faster transport properties must be developed.

Exploring the model DMEe K-ion chemistry further, where the electrolyte concentration overpotentials are not the limiting factor in determining rate, Fig. 4f shows the largest overpotential components are the negative and positive electrode concentration overpotentials with similar magnitudes. This reflects the transport limitations within the particles as the most significant factor limiting rate, though it is important to note the $\widetilde{D}_g$ and $\widetilde{D}_{KMF}$, utilised are likely underestimates as noted previously. Initially, the graphite concentration overpotential is most significant, yet, after SOC ~ 0.25 the KMF concentration overpotential becomes more dominant reflecting the KMF leaving the two-phase OCV plateau between $0.75 < x < 0.95$ (Fig. 3d), approaching the apparently undistorted phase at $0.4 < x < 0.75$ and thus the surface OCV increasing more rapidly. Outside of the KMF OCV plateau region the difference between the OCV at the KMF particle surface K$^+$ concentration, $U(c_{KMF,K^+}^s)$, compared to the OCV at the bulk KMF K$^+$ concentration, $U(c_{KMF,K^+})$, is larger, driving greater concentration overpotentials, $\eta_c$. Thus at this point during charging, with decreasing KMF K$^+$ concentration, $\eta_{c,KMF} = U(c_{KMF,K^+}) - U(c_{KMF,K^+}^s)$ becomes larger than $\eta_{c,g} = U(c_{g,K^+}) - U(c_{g,K^+}^s)$ causing the KMF concentration

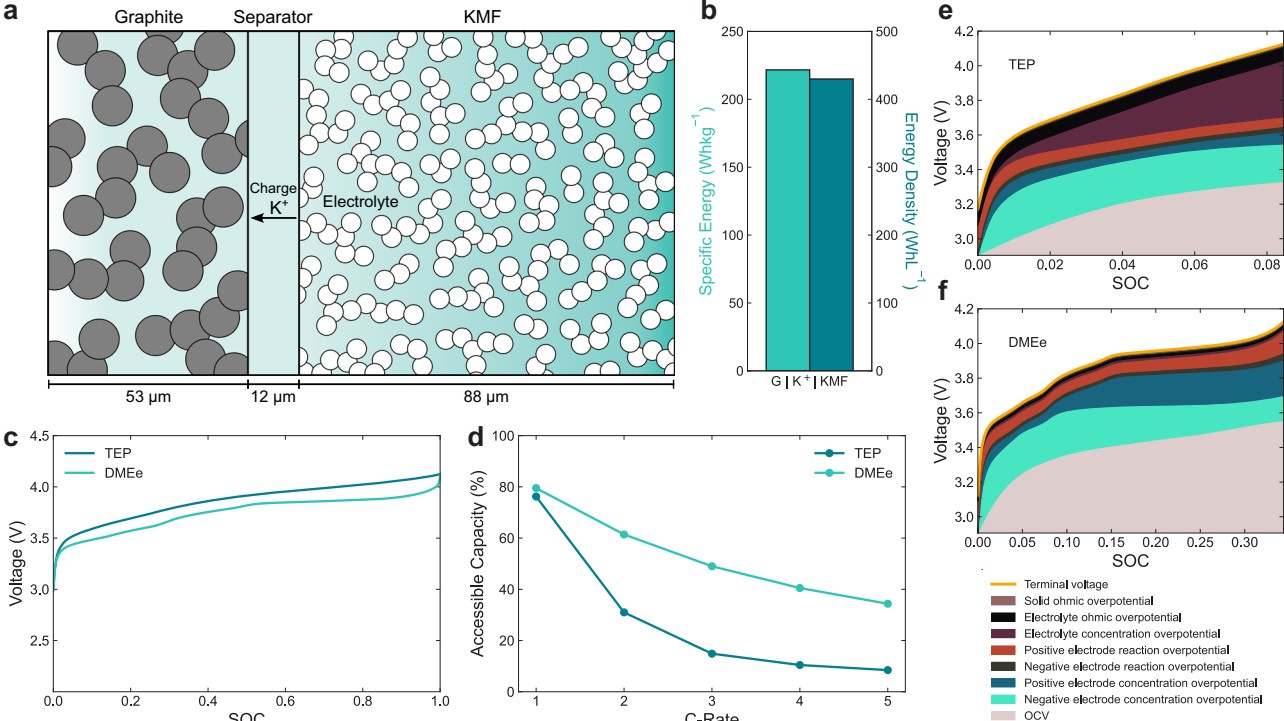

**Fig. 4 | K-ion full cell simulations.** Full-cell Doyle-Fuller-Newman (DFN) simulations of two K-ion cells with different electrolytes in a cell based on the commercial LG M50 cylindrical cell format. The state-of-the-art modelled K-ion cell is graphite (G) $\|$ K$_2$Mn[Fe(CN)$_6$] (KMF) with 2 m KFSI TEP electrolyte (denoted as TEP). The other cell is a hypothetical electrolyte case using the characterised electrolyte properties of the 2 m KFSI:DME[15] to simulate K-ion performance with an equivalent electrolyte (DMEe). The cell energy is 7.3 Wh. NP ratios were kept constant at 1.1

with constant electrode porosities, and properties for both chemistries. Modelled using PyBaMM[67] at 20 °C. (1 C = 1.91 mA cm$^{-2}$ for both). **a** Schematic of simulated K-ion cell. **b** Energy density and specific energy of the K-ion chemistry at the stack level using the stack-level model developed previously[5,70]. **c** Simulated galvanostatic profiles for the K-ion cells. **d** Accessible capacities at increasing C-rate. **e** Overpotential components during a 5 C charge for the TEP electrolyte. **f** Overpotential components during a 5 C charge for the DMEe electrolyte.

overpotentials to start dominating over the graphite. The positive electrode reaction overpotential is significantly larger than the negative reaction overpotential throughout the charge reflecting the slower $j_{0,KMF}$ than $j_{0,g}$, despite the smaller KMF particles and greater surface area for reaction. The electrolyte and solid ohmic overpotentials are very small contributions. An additional comparison using the electrolyte properties of the commercial Li-ion electrolyte LP57[72] (Supplementary Fig. 18 and Supplementary Note 4) showed negligible difference in rate performance with the DMEe electrolyte, showing the electrode material properties are rate limiting with a considerably slower transport electrolyte than DMEe.

Finally, since the LFP LIB is the Li-ion chemistry that K-ion would be competing with[5], an LFP model in the same LG M50 cylindrical cell was also simulated for comparison (Supplementary Note 5 and Supplementary Fig. 19). Again the total cell energies were matched by adjusting the electrode thicknesses, and the particle sizes for each chemistry's positve and negative electrode materials were kept constant between both models. KFSI and LiFSI in DME electrolytes were used for the K-ion and LFP Li-ion cell, respectively, for additional fair comparison and because both electrolytes were fully characterised over concentration in the same conditions previously[15]. Supplementary Fig. 19b shows the K-ion chemistry displays a comparable specific energy to an LFP chemistry (1% higher) though a significantly lower energy density (24% lower), due to the significantly lower bulk density of the KMF material compared to LFP (2.22 g cm$^{-3}$ vs. 3.45 g cm$^{-3}$[68], respectively (Supplementary Table 3)), again showing K-ion is competitive particularly where mass is a more critical constraint than space. Supplementary Fig. 19d shows the K-ion cell displays very similar rate performance to the LFP Li-ion cell (35% vs. 37% at 5 C, respectively), indicating K-ion is indeed competitive with LFP, should a

suitable electrolyte be developed. In fact, given the conservative K-ion median $\widetilde{D}$ assumptions, it is likely K-ion would exhibit superior rate performance to LFP from these results. The $\widetilde{D}_{LFP}$ has been tuned and estimated to experimental data[22], being ~ two to three orders of magnitude higher than an apparent median $\widetilde{D}_{LFP}$ from GITT or EIS results in other studies[33,73]. It is clear that the K-ion cell can experience larger overpotentials before reaching the upper cut-off voltage due to the KIB OCV profile with a more gradual increase compared to that of the LFP Li-ion cell (Supplementary Fig. 19e). One of the most significant reasons for this is the higher intercalation potential of K$^+$ compared with Li$^+$ into graphite (-0.3 V and 0.1 V, vs. K$^+$/Li$^+$, respectively)[5]. Supplementary Note 4 further explores the overpotential components of the chemistries.

Overall, the modelling draws two critical conclusions. First, an electrolyte with faster transport properties than the leading research KFSI:TEP electrolyte is required to realise the potential fast-charging capabilities of K-ion. Second, if an electrolyte is developed with transport comparable with KFSI:DME, the electrode material transport and kinetics become limiting in rate capability, not the electrolyte. To address the low $\widetilde{D}_{KMF}$ and $\widetilde{D}_g$ requires smaller particle sizes, making electrolyte stability even more important to maintain high coulombic efficiencies. The slower $j_{0,KMF}$ than $j_{0,g}$ reiterates the importance for researchers to focus on understanding the charge-transfer kinetics of the KMF positive electrode and improving them to match the faster kinetics at the graphite negative electrode.

## Discussion

In summary, we have accurately characterised the effective solid state diffusivities, $\widetilde{D}$, and exchange current densities, $j_0$, of the leading K-ion electrode materials, the graphite negative electrode and KMF positive

electrode. By synthesising highly homogeneous and non-agglomerated KMF particles we were able to more accurately determine the electrochemically active surface area $S$, one of the greatest sources of uncertainty in determining $\widetilde{D}$. We also applied the state-of-the-art Kang-Chueh GITT method and analysis to determine $\widetilde{D}$ more accurately. The median $\widetilde{D}_{KMF}$ is ~ four times lower than $\widetilde{D}_g$ and $j_{0,KMF}$ is ~ an order of magnitude lower then $j_{0,g}$, showing the KMF is the rate-limiting electrode material.

KFSI:TEP is currently one of the leading electrolytes for the K-ion research community using low loading in coin cells[42]. However, the full-cell K-ion modelling with realistic electrode thicknesses and loadings demonstrates this electrolyte is unsuitable for even moderately high charge rates using realistic electrode thicknesses and loading due to its slow transport. Therefore, an electrolyte with faster transport properties must be developed for K-ion batteries. Utilising the transport properties of KFSI:DME (DMEe) shows promising results for K-ion fast-charging. However, it is important to recognise that there is no current low-cost electrolyte which provides this performance and practical stability for KIBs. All current K-ion electrolytes suffer from impractically low initial coulombic efficiencies to be used in a commercial cell. The DMEe model has been developed to see what the rate capability of K-ion could be in realistic cell formats if a suitable electrolyte is developed which provides both a stable SEI for the graphite negative electrode and stability at the high operating voltages of the KMF positive electrode[5] with the fast transport properties of KFSI:DME[15]. This model should now support researchers in understanding the rate performance potential of K-ion, as well as understanding what the rate limitations are. The sluggish reaction kinetics and lower diffusivity in the KMF positive electrode requires nanosized particles to match the faster kinetics of the graphite negative electrode and achieve high rate performance, as is the case with LFP for LIBs. However, this also introduces challenges with significant passivation layer growth and electrolyte consumption unless the electrolyte is exceptionally stable. This is particularly challenging at the high operating voltages of the KMF. The inherently slow charge-transfer kinetics of the KMF must also be investigated, and optimisations explored. Nevertheless, this work suggests that the electrolyte is still a critical barrier to realising commercially viable KIBs. Achieving high stability and faster transport electrolytes must now be the priority in K-ion research. We hope this model will help guide the K-ion research community to develop optimised K-ion materials and electrolytes to realise fast-charging KIBs.

## Methods

### KMF synthesis

We synthesised a highly crystalline, monodisperse sample of $K_2Mn[Fe(CN)_6]$ to ensure accurate approximations could be made about surface area and diffusion length. We utilised a citrate-assisted co-precipitation in aqueous media. $MnSO_4$ (≥99%, Sigma-Aldrich, 0.5 mmol) was dissolved in an aqueous solution of potassium citrate (≥99%, Sigma-Aldrich, 1 M, 50 mL). $K_4Fe(CN)_6$ (≥99%, Sigma-Aldrich, 0.5 mmol) was dissolved in a separate aqueous solution of potassium citrate (≥99%, Sigma-Aldrich, 1 M, 50 mL). These solutions were added simultaneously, dropwise (2 mL min⁻¹), to a stirring round bottom flask containing a 100 mL solution of 1 M potassium citrate at 20 °C. The mixture was stirred for 24 h before the precipitate was isolated by centrifugation and washed with a 50:50 water/ethanol mixture in order to prevent the solid dispersing. The solid was dried in air at 70 °C overnight and then under vacuum at 70 °C.

### Synchrotron XRD

Synchrotron X-ray diffraction (XRD) measurements were performed on the I11 beamline of the Diamond Light Source, UK, operating with an X-ray wavelength of 0.824385 Å. The position-sensitive detector was used to collect diffraction patterns in capillary transmission geometry.

All refinements were carried out using the TOPAS-Academic software[74].

### SEM

Particles were dispersed in acetone or ethanol and then mounted on an aluminium stub. These particles were imaged using a Zeiss Merlin scanning electron microscope (SEM) equipped with a field emission gun, operated at an accelerating voltage of 3 kV or 10 kV and a probe current of 100 pA. The particle size distribution was determined by measuring at least 100 particles from SEM images using Feret's mean diameter and Fiji ImageJ software.

### Electrode preparation

The electrode loadings were low (~0.7 mg cm⁻² and 1.1 mg cm⁻² for graphite and KMF, respectively) with high porosity to minimise any electrolyte transport limitations, and ensure solid $D$ limitations. The graphite (SGP5, SEC Carbon) electrodes were composed of 92 wt% active material and 8 wt% sodium carboxymethyl cellulose (CMC) binder (Sigma-Aldrich) using highly purified deionised water as the solvent. The KMF electrodes were prepared using the mass ratio 7:2:1 active material:carbon:binder using Super P carbon additive (TIMCAL), polyvinylidene fluoride (PVDF) binder (Sigma-Aldrich) and 1-methyl-2-pyrrolidinone (NMP) solvent (99.5% anhydrous, Sigma-Aldrich). The electrodes were cast onto carbon-coated aluminium current collector (18 μm thick, MTI). All electrodes were dried first in air at room temperature for 24 h followed by vacuum drying at 100 °C for 24 h. The graphite electrodes were not calendered to ensure as high a porosity as possible. The KMF electrode was calendered until the top layer of the casting was just densified to ensure good electrical contact. Prior to use K metal (chunks, 98%, Sigma-Aldrich) was melted in an argon-filled glovebox, the impurity layer was removed, and the K was quenched into clean mineral oil, before being stored in hexane (95% anhydrous, Sigma-Aldrich)[15]. K counter electrodes were produced by rolling the K metal between two sheets of weighing paper (grade 2212, Whatman) to approximately 500 μm in thickness and punching it into 10 mm diameter electrodes with a wad punch. The surface of the K electrodes was polished with a plastic blade immediately before electrolyte addition.

### Electrochemistry

Prior to preparing electrolytes, potassium bis(fluorosulfonyl)imide (KFSI, 99.9%, Solvionic) was dried under vacuum at 100°C for at least 48 h and triethyl phosphate (TEP, 99.8%+, Sigma-Aldrich) was dried over potassium metal strips for at least 1 week. The water content of the electrolyte was measured by Karl Fischer titration and recorded to be below 5 ppm.

Three-electrode EL ECC-Ref cells (EL-CELL) were utilised for all Kang-Chueh GITT, PITT and EIS experiments using graphite or KMF working electrodes (10 mm diameter), K metal counter electrodes (10 mm diameter) and K metal reference electrodes. Glass microfiber separators were used (16 mm diameter, grade GF/F, Whatman). 600 μL of 2 m KFSI in TEP electrolyte was utilised for the Kang-Chueh GITT, PITT and EIS experiments. All Kang-Chueh GITT, PITT and EIS experiments were conducted in a Binder Oven at 20 °C (±0.3 K).

CR2032 coin cells were assembled to assess electrochemical performance using the same electrodes and separator as above, with 200 μL of 2 m KFSI in TEP. Aluminium-coated bases were used for the KMF cells due to the high operational voltages and steel cell parts for the graphite cells (MTI).

All electrochemical tests were carried out using a battery cycler (VMP3, Biologic). Electrochemical impedance spectroscopy (EIS) measurements were performed using a frequency response analyser (VMP3, Biologic), over the frequency range of 200 kHz–100 mHz (6 measurement points per decade) with an applied potentiostatic signal

of amplitude 10 mV. All EIS data was fitted using the Python package 'impedance.py'[75].

## Exchange current density

All EIS experiments were conducted in three-electrode ECC-Ref EL-cells to isolate the impedance and $R_{ct}$ of the working electrode. The cells were cycled once before conducting the EIS measurements over composition on the second cycle.

The exchange current density, $j_0$ [A cm$^{-2}$], varies with composition according to Eq. (3):

$$j_0 = k_0 \left(1 - \frac{c_s}{c_{s,max}}\right)^{\alpha_c} \left(\frac{c_s}{c_{s,max}}\right)^{\alpha_a} \left(\frac{c_e}{c_{e0}}\right)^{\alpha_c} \tag{3}$$

where $k_0$ [A cm$^{-2}$] is the reaction rate constant, $c_s$ [mol m$^{-3}$] is the potassium concentration at the particle surface, $c_{s,max}$ [mol m$^{-3}$] is the maximum potassium concentration in the active material, $c_e$ [mol m$^{-3}$] is the electrolyte concentration at the particle surface, $c_{e0}$ [mol m$^{-3}$] is the reference electrolyte concentration and $\alpha_a$ [−] and $\alpha_c$ [−] are the anodic and cathodic charge-transfer coefficients, respectively.

Full description of how $j_0$ is determined from $R_{ct}$ is as follows. The Butler-Volmer equation is first defined as:

$$j = j_0 \left(\exp\frac{\alpha_a F\eta}{RT} - \exp\frac{\alpha_c F\eta}{RT}\right) \tag{4}$$

Where $\eta$ [V] is the surface overpotential which is the difference between the electrolyte-electrode potential drop and the Nernst equilibrium potential $V_{eq}$, $\alpha_a$ and $\alpha_c$ are the anodic and cathodic direction charge-transfer coefficients, respectively. At small overpotentials around $\eta = 0$ the Butler-Volmer equation may be linearised:

$$j = j_0 \frac{F\eta}{RT} \tag{5}$$

As charge-transfer resistance, $R_{ct}$, is defined as:

$$\eta = jSR_{ct} \tag{6}$$

Combining Eqs. (5) and (6) produces an equation for $j_0$ with respect to $R_{ct}$:

$$j_0 = \frac{RT}{SFR_{ct}} \tag{7}$$

## Kang-Chueh GITT

All Kang-Chueh GITT experiments were conducted in three-electrode ECC-Ref EL-cells using a K metal reference electrode prepared using our K metal preparation protocol[15]. GITT experiments were conducted after one formation cycle at C/20. The Kang-Chueh GITT experiments were performed using C/20 15 min pulses and 2 h relaxations with application of current between to obtain a high-resolution OCV profile. The Kang-Chueh GITT pulse duration was selected to minimise finite-size effects[25,34]. At least two cells were utilised to determine the error. The OCV profile was determined from the Nernst potential points at the end of each relaxation step[25,26,34]. The Kang-Chueh GITT analysis uses relaxation-only analysis to avoid any overpotential errors according to Eq. (1)[25,34].

## Modelling

For the DFN full-cell modelling, the open-source battery simulation package Python Battery Mathematical Modelling (PyBaMM)[67] version 23.5 full-cell DFN model and CasADi numerical solver[76] was used. Full details of the parameters used are detailed in Supplementary Table 3. For determining the accessible capacity % at different C-rates, the

baseline performance was determined from a C/50 discharge (1 C = 1.91 mA cm$^{-2}$ for the K-ion cells and 2.24 mA cm$^{-2}$ for the LFP Li-ion cells). For the KFSI and LiFSI:DME electrolytes the empirical concentration-dependent functions characterised in our previous work were used[15]. The PyBaMM 'Chen2020' parameter set was used for the LG M50 cylindrical cell geometries and Li graphite material parameters[36]. For the K graphite electrode, the Li graphite electronic conductivity was used[36]. For the KMF positive electrode the Li-ion LFP electronic conductivity was used[22]. In order to achieve model convergence the graphite particle size of 1 μm was used; a commercial size (Supplementary Table 3) yet smaller than typical Li-ion and likely required in reality. The DFN model can fail to run when the characteristic relaxation time for solid-state diffusion, $\tau$ is too high ($\tau$ is related to both the solid-state diffusivity, $D$, and the particle radius, $r$, according to $\tau \propto \frac{r^2}{D}$). Therefore, although DFN models with larger graphite particle size have been reported for LIBs[20,36], these also utilise much larger diffusivities than measured here. It was therefore necessary to reduce the graphite particle size used in the model to aid convergence, which we believe is more appropriate than arbitrarily increasing the diffusivity. The maximum concentrations in each electrode [mol m$^{-3}$] were determined using[36]:

$$c_s^{max} = \frac{\rho z}{M} \tag{8}$$

where $\rho$ [kg m$^{-3}$] is the material crystal density and $M$ [kg mol$^{-1}$] is the molar mass of the active electrode material. The crystal density for the KMF positive electrode used in the modelling was derived from the unit cell volume determined from the refinement in combination with the KMF stoichiometric $M$.

## Data availability

All the experimental data used in this study are available in the Zenodo database under a Creative Commons Attribution 4.0 International License (https://doi.org/10.5281/zenodo.13122158)[77].

## Code availability

The Python codes used in this study are available in the Zenodo database under a Creative Commons Attribution 4.0 International License (https://doi.org/10.5281/zenodo.13122158)[77].

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

## Acknowledgements

This research was funded in whole, or in part, by the UKRI [EP/R010145/1]. For the purpose of Open Access, the author has applied a CC BY public copyright licence to any Author Accepted Manuscript version arising from this submission. The authors acknowledge the financial support of the Henry Royce Institute (through the above UK Engineering and Physical Sciences Research Council grant) for capital equipment, as well as use of characterisation facilities within the David Cockayne Centre for Electron Microscopy, Department of Materials, University of Oxford. S.D. appreciates the financial support from EPSRC and Shell. J.C. and A.L.G. gratefully acknowledge the E.R.C. for funding (Advanced Grant 788144) and the provision of a BAG allocation (CY25166) on the I11 beamline at the Diamond Light Source, U.K. B.J. is grateful for the support of the Clarendon Fund Scholarships. L.F.O. was supported by funding from EPSRC. We are grateful to Johannes Ihli for his ideas and thoughts, Isaac Capone for his blender modelling for Fig. 1, Stephen Hoy for some data fitting investigation, Robert Timms for his help with PyBaMM, Ruihe Li for his modelling feedback, Adam Lewis-Douglas for his modelling feedback, and Peter Klusener for his feedback.

## Author contributions

S.D. and M.P. conceptualised the study and designed the experiments; S.D. conducted all the GITT, PITT and exchange current density characterisation experiments and analysis; S.D. conducted the modelling; S.D. prepared the electrodes; J.C. synthesised the KMF and conducted the XRD experiments; J.C. and A.G. conducted the XRD analysis; M.S. and J.C. conducted the SEM experiments; S.D. conducted the SEM analysis; B.J. conducted supporting electrode materials investigation and produced the schematics; L.F.O. supported the EIS experiments and analysis, and blender modelling; J.Z. characterised the TEP electrolyte; Y.C. and K.S. conducted preliminary graphite investigation; M.S. supported the optimisation of the KMF electrode; S.D. wrote the original draft; S.D., B.J., J.C. and M.P. wrote, edited, and revised the manuscript; M.P. supervised the study and provided frequent input in the interpretation of all results.

## Competing interests

M.P. was scientific advisor to Project K Energy when the paper was originally submitted, however, he is no longer involved with the company. The remaining authors declare no competing interests.
