## [Peer Review File · Nature Communications]

Characterisation and Modelling of Potassium-Ion BatteriesREVIEWER COMMENTS

Reviewer #1 (Remarks to the Author):

This paper presents a full-cell Doyle-Fuller-Newman (DFN) model for a potassium-ion battery (KIB) based on a graphite anode and potassium manganese hexacyanoferrate ($\text{K}_2\text{Mn}[\text{Fe}(\text{CN})_6]$) cathode. Due to challenges with finding a suitable electrolyte, the authors could not present an experimental full-cell battery and therefore relied on experimental results from separate half-cells to develop the full cell model.

The highlight of this work is the characterization of graphite and $\text{K}_2\text{Mn}[\text{Fe}(\text{CN})_6]$ materials. The novelty lies in the kinetic rate parameters and solid-state diffusivity for the graphite anode and $\text{K}_2\text{Mn}[\text{Fe}(\text{CN})_6]$ cathode. Herein, the authors applied GITT and EIS techniques to determine the diffusion coefficients and kinetic rate constants respectively. The values thus obtained are comparable to other electrode materials and therefore within an expected range.

However, after reading the paper, the reviewer does not get the impression that the authors managed to identify an appropriate thesis statement. The goal/objective of the paper is not clear. My initial impression (after reading the introduction) was this paper wants to show the fast-charging potential of KIBs. However, this is not the case. In fact, neither the model nor the results support a claim that KIBs can have superior charging rates. Fast charging is an anode limited phenomena (because of limitations in overpotential at the anode when charging). For this the authors simply needed to show a half cell model of the KIB anode and compare that to a LIB anode. A DFN model is even an overkill in that case because the charging limitation is in the mass transport in the solid phases; a fact that the authors themselves state in this paper. In that case, a comparison of the kinetic rate and diffusion coefficient would suffice!

Overall, the reviewer does not see the contribution of the full-cell DFN model (without validation) to the paper. A half-cell model with validation would have been more impressive. In the previous work of the authors S. Dhir et al. Nat Commun 14, 3833 (2023), such a model had been presented and the reviewer does not think this model presented here offers more insight. If two half cell models had been validated and the results extrapolated to a full cell, that would have been more insightful. Therefore, I do not recommend publication of this work in Nature Communications.

Some specific comments:

1. Please try to avoid generalized statements which are not supported by references such as “The leading cathode material for potassium-ion batteries (KIBs), ...”. How is the material leading? Is it in use cases, in publications...?
2. Units are missing from the parameters presented in the main article. This makes it difficult to evaluate the equations.
3. The authors state that “The mean $j_{(0,g)}$ over the composition is $3.42 \times 10^{-5} \text{ A cm}^{-2}$. $j_{(0,g)}$ is similar to that found for Li graphite indicating similar charge-transfer reaction kinetics between Li^+ and K^+ and graphite.” How then can KIBs achieve faster charging?
4. The reviewer is not fully familiar with PyBaMM however it is known to be a P2D model. Therefore, it models a 1D cell and other additional components such as current collectors. My question is what do you gain by modelling an LG M50 cylindrical cell format? How is the result different for a pouch cell or single cell for example?
5. Parameters used for modelling are questionable.
 - 5a. Particle radius of 500 nm for graphite. This is unrealistic. Even the experimental results do not agree with this value.
 - 5b. The average Particle size for $\text{K}_2\text{Mn}[\text{Fe}(\text{CN})_6]$ in the experiments is around 1 μm yet the model uses 250 nm.
 - 5c. There is an Error in the KFSI:TEP salt diffusivity value/units in the SI table.
 - 5d. The model is simulated at 10 °C (283.15 K) while all the experiments are done at 20 °C.

6. Supplementary Fig. 19b confused me at first. It would be better to refer to gravimetric/specific energy density and volumetric energy density.

7. The authors state that “Nevertheless, this work again shows the electrolyte is the most critical barrier to realizing commercially viable fast-charging KIBs.” The reviewer does not agree with this conclusion because the hypothetical cell shows poor rate capability, and its limitations are in the electrodes where we see the highest overpotentials.

Reviewer #2 (Remarks to the Author):

This paper reports relevant work on K-ion batteries. This system has not been investigated in detail (i.e., in comparison to Li-ion systems), and the use of detailed experiments to inform the model are an important contribution. I have several additional comments about specific aspects of this work:

(1) One of their key conclusions (beginning of the Discussion) is that they have “accurately characterized the effective solid state diffusivities”. It would be useful if they could say more about the “error bars”, in light of the inherent complexity of battery electrodes. The authors do a good job of discussing the considerable challenges that are associated with measuring various quantities that are needed for data interpretation and modeling. In light of this, the error bars in Figs 16 and 17 (SI) seem rather small ?

(2) How do the model predictions at high C rates (Fig 4) compare with experiments ?

(3) Fast charging in Li-ion systems is limited by Li plating (on graphite). Is K plating an important problem ? Does the model predict this ?

(4) A minor point .. on page 11, the authors point to “two critical conclusions” that focus on the limitations imposed by the transport properties of the electrolyte. It seems like one could make this statement without their well calibrated model (i.e., by just knowing the electrolyte properties). I thought that the model predictions related to the kinetic limitations in the electrode materials were more interesting.

Response to the Reviewers' Comments

We would like to thank the reviewers for their insightful comments and appreciation for our work. Changes implemented to the original document are highlighted in red in the revised manuscript.

Reviewer #1

This paper presents a full-cell Doyle-Fuller-Newman (DFN) model for a potassium-ion battery (KIB) based on a graphite anode and potassium manganese hexacyanoferrate (K₂Mn[Fe(CN)₆]) cathode. Due to challenges with finding a suitable electrolyte, the authors could not present an experimental full-cell battery and therefore relied on experimental results from separate half-cells to develop the full cell model. The highlight of this work is the characterization of graphite and K₂Mn[Fe(CN)₆] materials. The novelty lies in the kinetic rate parameters and solid-state diffusivity for the graphite anode and K₂Mn[Fe(CN)₆] cathode. Herein, the authors applied GITT and EIS techniques to determine the diffusion coefficients and kinetic rate constants respectively. The values thus obtained are comparable to other electrode materials and therefore within an expected range.

We would like to thank the reviewer for showing appreciation for our work and for their valuable comments.

However, after reading the paper, the reviewer does not get the impression that the authors managed to identify an appropriate thesis statement. The goal/objective of the paper is not clear. My initial impression (after reading the introduction) was this paper wants to show the fast-charging potential of KIBs. However, this is not the case. In fact, neither the model nor the results support a claim that KIBs can have superior charging rates. Fast charging is an anode limited phenomena (because of limitations in overpotential at the anode when charging). For this the authors simply needed to show a half cell model of the KIB anode and compare that to a LIB anode. A DFN model is even an overkill in that case because the charging limitation is in the mass transport in the solid phases; a fact that the authors themselves state in this paper. In that case, a comparison of the kinetic rate and diffusion coefficient would suffice! Overall, the reviewer does not see the contribution of the full-cell DFN model (without validation) to the paper. A half-cell model with validation would have been more impressive. In the previous work of the authors S. Dhir et al. Nat Commun 14, 3833 (2023), such a model had been presented and the reviewer does not think this model presented here offers more insight. If two half cell models had been validated and the results extrapolated to a full cell, that would have been more insightful. Therefore, I do not recommend publication of this work in Nature Communications.

Upon careful review of the manuscript, we acknowledge the concerns raised by the reviewer and concur that the thesis statement lacks clarity in the introductory section. Our investigation aimed to assess the suitability of the KMF-graphite K-ion chemistry for fast charging. We have identified the sluggish reaction kinetics and lower diffusivity in the KMF cathode as significant impediments. However, DFN modelling of a realistic cell format underscore that the electrolyte poses the most critical barrier to achieving commercially viable fast-charging KIBs. We have modified the introduction of the manuscript to clarify this point.

The reviewer comment about fast charging as an anode limited phenomena is generally valid in commercial Li-ion batteries where lithium metal plating is limiting rate (due primarily to the low intercalation potential of 0.1 V vs. Li⁺/Li) but cannot be generalised to other chemistries. Transport, thermodynamic and kinetic properties of the active materials and electrolyte (all of which we have characterised) as well as electrode and cell design determine what the rate

determining process is. It would not be possible to accurately evaluate which process is limiting without a full-cell DFN model, and the different OCV profiles between K-ion and Li-ion electrode materials prevents a simple comparison of the kinetic rate and diffusion coefficients.

In Figure 4f we model a hypothetical electrolyte with DME properties (DMEe) and the rate is limited by K-ion depletion in the cathode as shown in Figure R1b. In contrast, when using the TEP electrolyte (Figure 4e), the rate is limited by the formation of significant K-ion concentration gradients within the electrolyte (and hence overpotentials) as shown in Figure R1f. Neither of these cases are limited by the anode.

Figure R1: K-ion particle and electrolyte concentrations from DFN simulations at 5C. Concentration of K^+ in the graphite anode particles within the particles (r dimension) and across the electrode (x dimension), and K^+ concentration in the electrolyte in the DMEe or TEP electrolyte K-ion cell after 5C charge once upper cut-off voltage is reached, corresponding to Fig. 4f and 4e, respectively. (a) K^+ concentration in the graphite anode particles in the DMEe cell. (b) K^+ concentration in the KMF cathode particles in the DMEe cell. (c) K^+ electrolyte concentration across the DMEe cell (d) K^+ concentration in the graphite anode particles in the TEP cell. (e) K^+ concentration in the KMF cathode particles in the TEP cell. (f) K^+ electrolyte concentration across the TEP cell.

Some specific comments:

1. Please try to avoid generalized statements which are not supported by references such as “The leading cathode material for potassium-ion batteries (KIBs), ...”. How is the material leading? Is it in use cases, in publications...?

We have now revised the main text and included references where needed.

2. Units are missing from the parameters presented in the main article. This makes it difficult to evaluate the equations.

We have now included units to the parameters in the equations presented.

3. The authors state that “The mean $j_{-}(0,g)$ over the composition is $3.42 \times 10^{-5} \text{ A cm}^{-2}$. $j_{-}(0,g)$ is similar to that found for Li graphite indicating similar charge-transfer reaction kinetics between Li^{+} and K^{+} and graphite.” How then can KIBs achieve faster charging?

j_0 is a function of electrolyte concentration, decreasing as the cation concentration at the anode interface decreases during charge. K-ion electrolytes can have better transport properties than the corresponding Li-ion electrolytes [[10.1038/s41467-023-39523-0](https://doi.org/10.1038/s41467-023-39523-0)], reducing the drop in j_0 for a given current density.

Potassium intercalation into graphite also occurs at 0.3 V vs. K^{+}/K (compared to lithium intercalation at 0.1 V vs. Li^{+}/Li), so larger overpotentials can be supported before potassium plating would occur, enabling higher current densities.

4. The reviewer is not fully familiar with PyBaMM however it is known to be a P2D model. Therefore, it models a 1D cell and other additional components such as current collectors. My question is what do you gain by modelling an LG M50 cylindrical cell format? How is the result different for a pouch cell or single cell for example?

An LG M50 cylindrical cell format was modelled because a comprehensive teardown analysis has been reported for this cell [[10.1149/1945-7111/ab9050](https://doi.org/10.1149/1945-7111/ab9050)], providing the key geometric parameters required for the model. This enabled us to ensure that we are modelling electrode geometries that would be relevant in commercial cylindrical cell formats. This therefore allows us to determine the processes that may limit the rate performance if this K-ion system was produced in a commercial format.

The results obtained from pouch cell or single cell models would be very similar providing the same electrode thicknesses and parameters are used.

5. Parameters used for modelling are questionable.

5a. Particle radius of 500 nm for graphite. This is unrealistic. Even the experimental results do not agree with this value.

As we mention in the Methods section, a graphite particle size of 1 μm (500 nm radius) was necessary for the model to converge. While this is smaller than typically used in Li-ion batteries (where radii from 5–10 μm are common [[10.1088/2516-1083/ac692c](https://doi.org/10.1088/2516-1083/ac692c)]), it is a particle size that is available commercially.

Importantly, the DFN models are not intending to replicate the characterisation experiments we performed but are instead modelling a full cell to predict potential performance in a commercial cell format.

The solid-state diffusivity characterisation experiments were performed with electrodes tailored to ensure accuracy, and these are not the same as would be used in a commercial cell. Larger graphite particles were used for the transport property characterisation to ensure that the relaxation of potential during the Kang-Chueh GITT was limited by solid-state diffusion

within these particles, enabling us to mitigate finite-size effects and improve the accuracy of the diffusivity value we determined.

5b. The average Particle size for $K_2[Mn[Fe(CN)_6]$ in the experiments is around $1\ \mu\text{m}$ yet the model uses 250 nm.

The KMF was synthesised with a large particle size specifically for GITT experiments to ensure solid-state diffusion limitation within the particles, as for the graphite. This size would not be appropriate use in commercial cells, hence a radius of 250 nm was used to match the typical size of lithium iron phosphate particles [[10.1016/j.electacta.2023.143341](https://doi.org/10.1016/j.electacta.2023.143341)].

5c. There is an Error in the KFSI:TEP salt diffusivity value/units in the SI table.

We thank the reviewer for pointing out this error. This has now been corrected.

5d. The model is simulated at $10\ ^\circ\text{C}$ (283.15 K) while all the experiments are done at $20\ ^\circ\text{C}$.

We thank the reviewer for pointing out this error. We confirm that all simulations were performed at 293.15 K ($20\ ^\circ\text{C}$) and we have corrected the text.

6. Supplementary Fig. 19b confused me at first. It would be better to refer to gravimetric/specific energy density and volumetric energy density.

Specific energy and energy density are common terms in the battery community and what we used in Figure 4b.

7. The authors state that “Nevertheless, this work again shows the electrolyte is the most critical barrier to realizing commercially viable fast-charging KIBs.” The reviewer does not agree with this conclusion because the hypothetical cell shows poor rate capability, and its limitations are in the electrodes where we see the highest overpotentials.

Figure 4e demonstrates that when using TEP as the electrolyte, the primary contributor to cell overpotential is the electrolyte itself. However, if an electrolyte with better transport properties, akin to the model system depicted in Figure 4f, were to be developed, the rate capability would be comparable to that of an equivalent LFP cell, as illustrated in Figure S19.

Acknowledging the reviewer's valid point, we recognize that this statement may downplay a crucial finding of the manuscript: that the kinetics at the PBA cathode are slower than previously suggested in the literature. Consequently, we have revised this sentence accordingly.

Reviewer #2

This paper reports relevant work on K-ion batteries. This system has not been investigated in detail (i.e., in comparison to Li-ion systems), and the use of detailed experiments to inform the model are an important contribution. I have several additional comments about specific aspects of this work:

(1) One of their key conclusions (beginning of the Discussion) is that they have “accurately characterized the effective solid state diffusivities”. It would be useful if they could say more about the “error bars”, in light of the inherent complexity of battery electrodes. The authors do a good job of discussing the considerable challenges that are associated with measuring

various quantities that are needed for data interpretation and modeling. In light of this, the error bars in Figs 16 and 17 (SI) seem rather small?

The error bars in Supplementary Figs. 16 and 17 represent the standard error in the mean values determined from two repeat cells.

From Eq. 1 in the main text, the main source of error that is not considered by this is any uncertainty in the electrochemically active surface area, S . According to Fig. 2c, 3c and Supplementary Fig. 2, we estimate the uncertainty in particle dimensions to be on the order of 10%, resulting in an uncertainty in S of approximately 20% which is lower than the cell to cell variation.

$$\frac{\sigma_D}{D} \approx \sqrt{\left(2\frac{\sigma_S}{S}\right)^2 + \left(\frac{\sigma_{f(V)}}{f(V)}\right)^2} = \sqrt{0.16 + \left(\frac{\sigma_{f(V)}}{f(V)}\right)^2}$$

(2) How do the model predictions at high C rates (Fig 4) compare with experiments?

Unfortunately, the K-ion research community is not in a position to assemble reproducible and therefore meaningful full cells due to a combination of the poor performance of the existing electrolytes, as well as the need to reengineer inactive components such as binder, conductive additives, current collectors, and separators. There are a couple of start-ups that are now working on commercialising this chemistry (Group 1 and Project K Energy). We hope to be able to validate our model against optimised prototypes soon.

(3) Fast charging in Li-ion systems is limited by Li plating (on graphite). Is K plating an important problem? Does the model predict this?

Yes, K plating on charge is an important problem that must be avoided in K-ion cells. Conditions that could lead to K plating are prevented in the model by the introduction of an upper cut-off voltage of 4.125 V, which limits the potential of the graphite anode to be > 0 V vs. K^+/K . This was the limit used when evaluating the accessible capacities and overpotential contributions in Fig. 4d and e, respectively.

It should also be noted that a potential advantage of the K-ion system is that K^+ intercalation into graphite occurs at 0.3 V vs. K^+/K (compared to 0.1 V vs. Li^+/Li for Li^+ intercalation), so much larger overpotentials can be supported before metal plating could occur.

(4) A minor point .. on page 11, the authors point to “two critical conclusions” that focus on the limitations imposed by the transport properties of the electrolyte. It seems like one could make this statement without their well calibrated model (i.e., by just knowing the electrolyte properties). I thought that the model predictions related to the kinetic limitations in the electrode materials were more interesting.

Transport, thermodynamic and kinetic properties of the active materials and electrolyte (all of which we have characterised) as well as electrode and cell design determine what the rate determining process is. Without a full-cell DFN model it's not possible to make an accurate prediction of the electrolyte properties necessary for rate capability to be limited by solid-state diffusivity within the active materials.

REVIEWER COMMENTS

Reviewer #1 (Remarks to the Author):

I appreciate the work done to revise the manuscript as well as the responses to my previous comments. In particular the improved thesis statement and conclusions are noted. Overall , I am pleased with the experimental section of the work but still not convinced with the modelling section. In particular, the issue of small particle sizes used with the only explanation being "in order to facilitate model convergence". P2D models with larger particle sizes exist in literature so this definitely not an inherent characteristic of the P2D model. Another issue is the claim that the model depicts a "commercial cell format". It is simply not possible to have cell geometric features included at this stage of the model. At best the model can show performances at coin cell level; there is no need to make such big claims.

Reviewer #2 (Remarks to the Author):

The authors have provided reasonable responses to my original comments. This work will be of interest to the community, and I believe that publication is warranted.

In reviewing the details, I think that the authors have cut some corners. Two notable examples are that diffusivities were obtained from only two cells (more cells would be preferable). Also, more effort could perhaps have been devoted to improving convergence issues, to run calculations for larger graphite particle sizes.

Response to the Reviewers' Comments

We would like to thank the reviewers for their insightful comments and appreciation for our work. Changes implemented to the original document are highlighted in red in the revised manuscript.

Reviewer #1

I appreciate the work done to revise the manuscript as well as the responses to my previous comments. In particular the improved thesis statement and conclusions are noted. Overall, I am pleased with the experimental section of the work but still not convinced with the modelling section.

We would like to thank the reviewer for their comments, which have helped improve the manuscript.

In particular, the issue of small particle sizes used with the only explanation being "in order to facilitate model convergence". P2D models with larger particle sizes exist in literature so this definitely not an inherent characteristic of the P2D model.

It is well known that the DFN model can fail to run when the characteristic relaxation time for solid-state diffusion, τ , is too high. τ is related to both the solid-state diffusivity, D , and the particle radius, r , according to $\tau \propto r^2 D^{-1}$. The reviewer is correct that models with larger graphite particle sizes have been reported for lithium-ion batteries, but the diffusivities are also significantly higher. In fact, measured D values are commonly scaled by several orders of magnitude for the P2D model to run with these large particle sizes. For example, both Chen et al. and O'Regan et al. utilised a radius of 5.86 μm and diffusivities between 10^{-14} and $10^{-13} \text{ m}^2 \text{ s}^{-1}$ [[10.1149/1945-7111/ab9050](https://doi.org/10.1149/1945-7111/ab9050), [10.1016/j.electacta.2022.140700](https://doi.org/10.1016/j.electacta.2022.140700)]. In this work we measured a graphite solid-state diffusivity of $2.32 \times 10^{-17} \text{ m}^2 \text{ s}^{-1}$, resulting in a characteristic relaxation time that is between 3 and 4 orders of magnitude higher than in these other reports if we use the same particle radius, resulting in issues with convergence of the DFN model.

The conventional solution to overcome solver errors, as highlighted in a recent article by Offer et al. [[10.1016/j.jpowsour.2024.234184](https://doi.org/10.1016/j.jpowsour.2024.234184)], is to either increase the diffusivity or decrease the current density. However, we have made significant efforts to accurately measure diffusivities and believe that a more appropriate approach is to reduce particle size to a value that has been experimentally demonstrated to be achievable, rather than arbitrarily increasing diffusivity to aid convergence. The graphite particle radius of 500 nm used in the model was the largest size that could run (even when reducing the current density to C/50), and we believe this model still provides valuable insight into the electrode and electrolyte properties limiting rate performance in potassium-ion batteries.

Another issue is the claim that the model depicts a "commercial cell format". It is simply not possible to have cell geometric features included at this stage of the model. At best the model can show performances at coin cell level; there is no need to make such big claims.

In this context, "commercial format" simply means that we used parameters from a comprehensive teardown of a commercially available LG M50 lithium-ion cell format [[10.1149/1945-7111/ab9050](https://doi.org/10.1149/1945-7111/ab9050)], ensuring realistic electrode thicknesses and loadings could be used in the model. We have modified the text to avoid any confusion.

The abstract now reads:

Finally, we present the first Doyle-Fuller-Newman model of a KIB full cell with realistic geometry and loadings, identifying the critical materials properties that limit their rate capability.

The introduction now reads:

Finally, we present the first Doyle-Fuller-Newman model of a KIB full cell in a hypothetical cell based on the commercial LG M50 cylindrical cell format, enabling us to identify the critical limitations in realising fast-charging KIBs.

The “Full-cell Potassium-ion Modelling” section now reads:

To understand the potential of KIBs for fast-charging, we developed a KIB full-cell DFN model in a hypothetical cell based on the commercial LG M50 cylindrical cell format, ensuring realistic electrode thicknesses, parameters and loadings were used.

The caption of Figure 4 now reads:

Full-cell Doyle-Fuller-Newman (DFN) simulations of two K-ion cells with different electrolytes in a cell based on the commercial LG M50 cylindrical cell format.

The “Discussion” section now reads:

However, the full-cell K-ion modelling with realistic electrode thicknesses and loadings demonstrates this electrolyte is unsuitable for even moderately high charge rates using realistic electrode thicknesses and loading due to its slow transport.

Reviewer #2

The authors have provided reasonable responses to my original comments. This work will be of interest to the community, and I believe that publication is warranted.

We would like to thank the reviewer for their comments, which have helped improve the manuscript.

In reviewing the details, I think that the authors have cut some corners. Two notable examples are that diffusivities were obtained from only two cells (more cells would be preferable).

Properly assembling three-electrode cells for GITT measurements is generally known to be tricky. This task becomes even more challenging when working with highly reactive potassium metal. Additionally, each measurement takes several weeks to complete. The two measurements reported here are in good agreement, and we believe they accurately represent the measured diffusivities.

Also, more effort could perhaps have been devoted to improving convergence issues, to run calculations for larger graphite particle sizes.

Please refer to the response to the identical comment made by Reviewer 1 above.